# ATM Pathway Is Essential for HPV–Positive Human Cervical Cancer-Derived Cell Lines Viability and Proliferation

**DOI:** 10.3390/pathogens11060637

**Published:** 2022-06-01

**Authors:** Walason Abjaude, Bruna Prati, Veridiana Munford, Aline Montenegro, Vanesca Lino, Suellen Herbster, Tatiana Rabachini, Lara Termini, Carlos Frederico Martins Menck, Enrique Boccardo

**Affiliations:** 1Laboratory of Oncovirology, Department of Microbiology, Instituto de Ciências Biomédicas, Universidade de São Paulo, Sao Paulo 05508-900, Brazil; bijaude@gmail.com (W.A.); brunaprati@yahoo.com.br (B.P.); montenegro.aline@hotmail.com (A.M.); vanesca_lino@usp.br (V.L.); sueherbster@usp.br (S.H.); 2Laboratory of DNA Repair, Department of Microbiology, Instituto de Ciências Biomédicas, Universidade de São Paulo, Sao Paulo 05508-000, Brazil; vmunford@usp.br (V.M.); cfmmenck@usp.br (C.F.M.M.); 3Institute of Pharmacology, Inselspital, INO-F, University of Bern, CH-3010 Bern, Switzerland; tatianarabachini@hotmail.com; 4Centro de Investigação Translacional em Oncologia, Instituto do Câncer do Estado de São Paulo, Hospital das Clínicas da Faculdade de Medicina da Universidade de São Paulo, Sao Paulo 01246-000, Brazil; terminilara@gmail.com

**Keywords:** HPV, DNA repair, synthetic lethality, ATM, CHK2, BRCA1

## Abstract

Infection with some mucosal human papillomavirus (HPV) types is the etiological cause of cervical cancer and of a significant fraction of vaginal, vulvar, anal, penile, and head and neck carcinomas. DNA repair machinery is essential for both HPV replication and tumor cells survival suggesting that cellular DNA repair machinery may play a dual role in HPV biology and pathogenesis. Here, we silenced genes involved in DNA Repair pathways to identify genes that are essential for the survival of HPV-transformed cells. We identified that inhibition of the ATM/CHK2/BRCA1 axis selectively affects the proliferation of cervical cancer-derived cell lines, without altering normal primary human keratinocytes (PHK) growth. Silencing or chemical inhibition of ATM/CHK2 reduced the clonogenic and proliferative capacity of cervical cancer-derived cells. Using PHK transduced with HPV16 oncogenes we observed that the effect of ATM/CHK2 silencing depends on the expression of the oncogene E6 and on its ability to induce p53 degradation. Our results show that inhibition of components of the ATM/CHK2 signaling axis reduces p53-deficient cells proliferation potential, suggesting the existence of a synthetic lethal association between CHK2 and p53. Altogether, we present evidence that synthetic lethality using ATM/CHK2 inhibitors can be exploited to treat cervical cancer and other HPV-associated tumors.

## 1. Introduction

Human Papillomaviruses (HPV) are small, non-enveloped, epitheliotropic, double-stranded DNA viruses commonly detected in anogenital epithelia, oral cavity mucosa, and skin. Mucosal HPV types can be classified as low- or high-oncogenic risk according to the main pathologies caused by these agents. Infection with low-risk HPV types is associated with the development of genital warts while high-risk types, particularly HPV type 16 (HPV16) and type 18 (HPV18), are associated with cervical cancer, the fourth most common cancer in women worldwide [1,2]. Infection with HPV16 has also been associated with a significant fraction of anal, vulvar, vaginal, penile, and oropharynx carcinomas. At present, three HPV vaccines have been approved for use in women and men in several countries. These vaccines effectively reduce the risk of infection by the HPV types included in their formulations and will undoubtedly contribute to the reduction of HPV-related diseases burden in the future. However, these vaccines are prophylactic and not effective for treating established infections or tumors. Therefore, the development of therapeutic strategies targeting HPV-related lesions is still a challenge.

Genomic instability is a hallmark of HPV infection [3,4,5]. During HPV infection, several proteins involved in DNA repair are recruited to viral replication centers. These cellular factors are crucial for viral genome amplification and co-localize with viral replication foci in the nucleus of the host cell [6,7]. Interestingly, several reports have shown that HPV early proteins may promote genome instability. For instance, it was observed that the combined expression of E1 and E2 proteins caused the increase of reactive oxygen species (ROS) levels and the accumulation of DNA damage [8]. Moreover, results from different groups showed that sustained E1 and E2 activity may induce the accumulation of DNA lesions through the replication of episomal and integrated viral genomes [9,10,11]. The detailed description of the mechanisms involved are beyond the scope of our study and the reader is referred to some excellent reviews [12,13]. High-risk HPV express two viral oncogenes, E6 and E7, which sustained expression is required during malignant transformation. These oncoproteins target several host factors to promote a cellular milieu permissive for efficient viral replication. For instance, E6 and E7 target tumor suppressors p53 and pRB, respectively, and induce their degradation to promote unscheduled cell proliferation while avoiding apoptosis [14,15]. It has been observed that expression of the HPV oncogenes induces abnormal centrosome numbers, aberrant mitotic spindle pole formation and modulates the cellular response to DNA damage [6,11,16,17,18,19]. Interestingly, HPV16 E6 protein expression has been associated to the loss of p53R2 activation in response to DNA damage and p53-mediated oxidative stress [20] while fibroblasts expressing HPV16 E6 and/or E7 are deficient in nucleotide excision repair (NER) [21]. Moreover, the expression of both oncoproteins is associated with the occurrence of DNA breaks facilitating the integration of exogenous DNA into the genome of the host cell [19,22].

These observations suggest that DNA damage repair machinery may play a dual role in HPV biology and pathogenesis. First, it is important for efficient viral genome amplification during HPV infection. Second, the presence of efficient DNA repair mechanisms may be crucial to maintaining the minimal genome stability necessary for tumor cells to survive and proliferate [13]. Considering this, we hypothesize that HPV-transformed cells may be highly dependent on specific DNA damage repair pathways. To test this hypothesis, we transduced cervical cancer-derived cell lines (HeLa and SiHa) and primary human keratinocytes (PHK) with specific shRNA targeting 116 genes involved in DNA Repair pathways. This approach allowed us to identify genes essential for HPV-transformed cells survival, but not required for normal cells viability. We show that the down-regulation of ATM, BRCA1, and CHEK2 genes expression selectively affected HPV-positive cervical cancer-derived cells proliferation/viability. Moreover, we show that gene silencing or chemical inhibition of the kinases encoded by ATM and CHEK2 reduced the clonogenic and proliferative capacity of cervical cancer-derived cells. We provide evidence that this effect is paralleled by an increase in the number of hypodiploid cells. Experiments conducted using PHK transduced with HPV16 oncogenes show that the effect of ATM, CHEK2 and BRCA1 silencing depended on the expression of E6 oncogene and on its ability to induce p53 degradation. Importantly, we provide evidence that treatment of p53-deficient cells from different origins with ATM/CHK2 inhibitors downregulates cell proliferation. Our results show that inhibition of components of the ATM signaling axis reduces p53-deficient cells proliferation potential, suggesting the existence of a synthetic lethal association between CHEK2 and p53. These preclinical observations may prove relevant for the development of alternative therapeutic strategies to HPV-positive tumors.

## 2. Results

### 2.1. ATM, BRCA1 and CHEK2 Are Essential for Proliferation/Viability of HPV-Positive Cervical Cancer-Derived Cell Lines

Cellular DNA repair machinery plays a critical role in HPV life cycle [6,7]. Considering this, we aimed to identify which genes involved in DNA repair pathways are required to sustain cervical cancer-derived cell lines proliferation and viability. High-risk HPV positive cell lines HeLa and SiHa were transduced with a commercial lentiviral-based library of shRNA targeting one hundred and sixteen genes involved in DNA damage repair signaling pathways. The library is composed of at least five different shRNA clones for each gene to maximize the possibility of efficient gene silencing. Five days after infection, cell viability was determined by Alamar Blue assay. The values of Alamar Blue reduction for each cell line transduced with each shRNA clone were normalized to cells transduced with a Scrambled control shRNA. Genes were considered as candidates when at least one shRNA clone was able to reduce proliferation/viability more than 30% in both cervical cancer cell lines (Appendix A). We identified 39 shRNA clones targeting 23 different genes that met this criterion (Figure 1A). We then sought to determine which of these genes were essential for cervical cancer cell lines viability, but were not required for normal primary human keratinocytes (PHK) survival. After transducing PHK, HeLa, and SiHa cells with shRNA targeting the 23 candidate genes, we identified three members of the ATM/CHEK2 signaling pathway (ATM, BRCA1, and CHEK2) which silencing reduced HeLa and SiHa proliferation without affecting normal PHKs (Figure 1B). A similar result was observed upon HMGB1 silencing and is reported elsewhere [23]. The silencing of all other 19 genes reduced the viability of both normal PHK and cervical cancer-derived cell lines (Appendix A).

### 2.2. Gene Silencing or Chemical Inhibition of ATM and CHEK2 Reduces the Proliferation and Clonogenic Capacity of Cervical Cancer-Derived Cells Lines

To further demonstrate the dependence of cervical cancer-derived cells lines on the identified genes, we analyzed the effect of gene silencing using different shRNA on PHK, HeLa and SiHa cells proliferation and clonogenic potential. Immunoblots against ATM and CHK2 confirmed that two shRNA targeting ATM and three shRNA targeting CHEK2 were able to strongly repress the expression of the target protein in the three cell lines (HeLa, SiHa, and PHK) (Figure 2A). Growth curves indicate that the tumor-derived cell lines expressing different shRNAs against ATM and CHEK2 had their proliferation potential dramatically reduced, but that silencing of these genes did not affect the proliferation of PHK (Figure 2B). Tumor cells (HeLa and SiHa) also exhibit reduced colony formation and anchorage-independent growth potential upon ATM and CHEK2 silencing when compared to cells expressing only scrambled shRNA (Figure 2C,D; Appendix A). Similar results were observed in cells expressing shRNAs against BRCA1 (Appendix A). These results indicate that ATM/CHK2 signaling pathway is important for HPV-positive cells proliferation and may play a role in sustaining cervical cancer-derived cells transformed phenotype.

### 2.3. Chemical Inhibition ATM Reduces Growth Potential and Viability of Cervical Cancer-Derived Cell Lines

ATM protein kinase is activated by DNA damage. Once activated it triggers phosphorylation of its downstream targets leading to cell cycle arrest, DNA repair, or apoptosis. Among its downstream targets there are several tumor suppressors proteins including CHK2, BRCA1, p53, and histone H2AX [24]. ATM inhibition can be achieved by treatment with caffeine (an ATM/ATR inhibitor) or, more specifically, by KU-55933 (an ATM specific inhibitor; Appendix A), not interfering with endogenous levels of this kinase [24]. Therefore, in order to confirm if ATM is critical for maintaining cervical cancer-derived cell lines proliferation, we treated HeLa and SiHa cell lines, as well as normal PHK, with 2 mM caffeine and 10 μM KU-55933. Our results show that, upon ATM inhibition with caffeine or KU-55933, proliferation is strongly impaired in both cervical cancer cell lines, but not in PHK (Figure 3A,B). Interestingly, HeLa and SiHa cells proliferation was also impaired by treatment with 100 μM CHK2 inhibitor while normal PHKs were only slightly affected by the treatment (Figure 3C; Appendix A). To further investigate the effect of ATM and CHK2 inhibition on HPV-transformed cells, we performed a cell cycle analysis by flow cytometry. We found that the treatment of SiHa and HeLa cells with caffeine or KU-55933 decreased cell viability as demonstrated by the accumulation of cells in sub-G1. The same was not observed or was observed to a much lesser extend for PHK (Figure 3D). Viability dropped even more in HeLa and SiHa cells treated with a CHK2-specific inhibitor. Once more, the effect of this inhibition was much less apparent in PHK. Altogether, these data indicate that HPV-positive cervical carcinoma cell lines are strongly dependent on ATM/CHK2 pathway for their survival.

### 2.4. Sensitivity of Cervical Cancer-Derived Cell Lines to ATM and CHK2 Inhibition Depends on the Ability of HPV E6 to Induce p53 Degradation

Sustained expression of high-risk HPV oncogenes E6 and E7 is necessary to maintain the transformed phenotype of cervical cancer-derived cell lines [14,25]. Together, E6 and E7 proteins are able to interfere with cellular proliferation and apoptosis, especially due to their ability to induce p53 and pRb degradation, respectively [15]. Considering this, we aimed to investigate if the expression of these oncoproteins could play a role in the sensitivity of cervical cancer-derived cell lines to ATM and CHK2 inhibition. Our results indicate that PHK expressing HPV–16 oncoproteins were differentially affected by ATM and CHEK2 gene silencing (Appendix A). PHK expressing HPV–16 E6 and E7 or only HPV–16 E6 had their proliferation remarkably affected by ATM and CHEK2 downregulation. Control PHK and PHK expressing only HPV–16 E7 were not affected by gene silencing (Figure 4A,B). Similar results were found after depletion of BRCA1 in PHK (Appendix A). The inhibition of ATM kinase with caffeine or KU-55933, as well as CHK2 inhibition, also led to a decrease in proliferation of cells expressing HPV–16 E6 (Figure 4C–E). Moreover, CHK2 inhibition also caused a decrease in proliferation of cells expressing HPV–16 E7 alone, although to a lesser extent than that observed in cells expressing the oncogene E6. Interestingly, the decrease in proliferation observed in PHK expressing E6 was much less effective in PHK transduced with HPV–16 E6^8S9A10T^ unable to induce p53 degradation (16ΔE6). Similarly, PHK expressing shRNA against the ubiquitin ligase responsible for p53 degradation in the presence of E6, E6-associated protein (E6-AP), did not have proliferation strongly affected by ATM or CHK2 inhibition (Figure 4C–E; Appendix A) [26]. These results indicate that the sensitivity of cervical cancer-derived cell lines to ATM and CHK2 inhibition is dependent, at least in part, on the ability of HPV E6 to induce p53 degradation.

### 2.5. ATM/CHEK2 Inhibition Potentiates the Effect of Doxorubicin and Cisplatin in a p53-Dependent Manner

Doxorubicin is a chemotherapeutic drug used to treat several types of cancer. It interacts with DNA and stops the progression of topoisomerase II, blocking cell division by inhibiting DNA duplication. It is also known to induce ATM-dependent phosphorylation of several downstream targets [27,28]. Cisplatin is also used as a chemotherapeutic agent and crosslinks DNA impairing cell division by mitosis [29]. DNA damage induced by both drugs elicits DNA damage repair machinery blocking proliferation or inducing apoptosis when repair proves impossible. To verify the cervical cancer cell lines dependency on ATM/CHK2 signaling pathway upon DNA damage we treated HeLa and SiHa cells with doxorubicin and/or cisplatin. We observed that both cell lines had proliferation/viability slightly decreased by the treatment with doxorubicin (1 μM) or cisplatin (1 μM). This decrease was not significant and was not intensified by the combination of both drugs (Figure 5A). As observed before, ATM inhibition led to a drop in cell viability. Moreover, the combined treatment with ATM inhibitors (KU-55933 or caffeine) with doxorubicin promoted a stronger drop of (~60%) in cell viability. Interestingly, the combination of the specific ATM inhibitor KU-55933 with cisplatin did not affect cell survival. Only caffeine, an inhibitor of both ATM and ATR, promoted a drop in cell viability when combined with cisplatin. Surprisingly, the combination of KU-55933 with doxorubicin and cisplatin led to a significant drop in viability, with only 20% of cells surviving after treatment. We next investigated the response of PHK expressing HPV-16 E6 or E7 oncoproteins to the combined treatment of ATM inhibitors with doxorubicin and cisplatin. The results indicate that ATM/CHK2 inhibition potentiates the effect of doxorubicin and cisplatin in cells expressing HPV-16 E6 (Figure 5B). Again, the integrity of E6 sequences required to mediate p53 degradation by E6 seems to play a major role in this phenomenon. This is supported by the fact that PHK expressing the mutant form E6^8S9A10T^, unable to bind p53, or shRNA against E6-AP were only mildly affected by the combined treatment of doxorubicin and cisplatin with ATM inhibitors (Figure 5C; Appendix A). The specific inhibition of CHK2 also turned PHK expressing E6 extremely sensitive to doxorubicin or cisplatin treatment (Figure 5D). Moreover, p53 inactivation also played an important role in sensitizing PHK expressing E6 to the combined inhibition of CHK2 with doxorubicin and/or cisplatin (Figure 5E). Finally, we investigated if p53 inactivation, by deletion or mutation, could sensitize cells to ATM inhibition. Therefore, we treated HaCat, (immortalized keratinocyte; p53mutH179Y/R282W) and C33 (HPV-negative cervical cancer cell line; p53R273C) with ATM inhibitors in combination with doxorubicin and/or cisplatin. Our results show that HaCat immortalized keratinocytes and C33 cells are as sensitive to ATM inhibition as they are to doxorubicin, with proliferation decreasing to 60% compared to the untreated control (Figure 5F). Cisplatin alone did not affect HaCat viability. Moreover, in HaCaT we did not observe additive or potentializing effect of ATM inhibitor in combination with doxorubicin and/or cisplatin. On the other hand, we observed that C33 cells, in which wild type p53 activity is not detected due to a R273C mutation, had viability dropping to 20% after combined treatment of ATM inhibitor with doxorubicin and cisplatin. Altogether, our results indicate that p53 deletion or inactivation by HPV-16E6 expression makes cells more prone to die when ATM is inhibited in combination with drugs that induce DNA damage or affect cell division, such as doxorubicin and cisplatin.

### 2.6. CHEK2 and BRCA1 Expression Is Higher in Precursor Lesions and Cervical Cancer

Finally, we evaluated the mRNA expression of ATM, CHEK2, and BRCA1 genes on clinical cervical tissues from GEO public platform. We retrieved normalized mRNA expression results and clinical data from four datasets that included normal cervix tissue, cervical intraepithelial neoplasias (CIN) of different grades and cervical cancer. The four datasets presented different experimental designs and criteria for normal cervix tissue, as GSE39001 and GSE52904 included HPV negative normal cervix samples from patients undergoing hysterectomy, while GSE63514 and GSE138080 considered normal-HPV positive epithelium as controls [30,31,32,33]. Overall, we observed higher mRNA levels of CHEK2 and BRCA1 genes in CIN and cervical cancer samples when compared to normal cervix tissue (Figure 6A–H). The results presented here show that the increased expression of CHEK2 and BRCA1 is a common event in the natural history of cervical cancer. It is noteworthy that ATM mRNA expression was not altered in any of the datasets analyzed in this study.

## 3. Discussion

In HPV-transformed cells, the oncoproteins E6 and E7 are constitutively expressed and impair multiple DNA repair pathways such as those regulated by p53 and pRb. This promotes cell cycle progression, avoids apoptosis, and consequently causes genomic instability [19,34,35,36]. Due to the redundant nature of these pathways, many tumors are able to efficiently repair different DNA lesions and survive in the presence of partially damaged DNA repair machinery. This fact makes the tumor completely dependent to alternative ways of DNA repair that can be exploited to develop specific toxicity for the tumor, contributing to the implementation of new therapeutic strategies. Based in these insights we screened several DNA repair pathways, by gene RNA silencing, which could be required for survival of HPV transformed cell during the tumor progression. The main idea was to establish potential synthetic lethality strategies to target HPV-associated tumors. Synthetic lethality describes a cellular condition in which two or more non-allelic and non-essential mutations, which are not lethal on their own, become deadly when present within the same cell. The synthetic lethality can also occur when a gene is inactivated by viral protein expression and, another gene, is inactivated by specific inhibitors. This principle has been used to determine the effect of synthetic lethality between p53 and PAK3 and SGK2 kinases in HPV transformed cells. In the presence of the HPV E6 oncoprotein these cells required these kinases to survive [37]. In addition, many studies have been conducted with the systematically silencing of tumor suppressors or oncogenes to exploit synthetic lethality in the study of cancer [38,39,40].

As a result of our screening, we identified that the signaling pathway controlled by ATM, CHEK2, and possibly BRCA1 is important to maintain the proliferation and viability of HPV-transformed cells. Moreover, we showed that down-regulation of these pathways affected the oncogenic potential of these cells in vitro. Additional experiments showed that the effect observed depended on the expression of HPV16 E6 and on the integrity of sequences required to mediate p53 degradation. In fact, we demonstrated that the effect of ATM/CHEK2 silencing or chemical inhibition of the proteins encoded by these genes was not observed in cells expressing an HPV-16 E6 mutant unable to degrade p53. This was also true in cells where E6-AP, a factor required for E6-mediated p53 degradation, was silenced by shRNA. The existence of a synthetic lethal relationship between ATM and p53 has been reported before in different systems and has been revised elsewhere [41,42]. Our results further extend these observations and support the existence of synthetic lethality between p53 and CHEK2, and possibly BRCA1 in HPV-transformed cells. Being downstream in ATM signaling, our observation brings an interesting piece of information that, hopefully, could contribute to fully understand the molecular mechanisms underlying this phenomenon.

Active HPV DNA amplification requires the phosphorylation of several ATM substrates, such as BRCA1, RAD51, and CHK2 [7,43,44]. Furthermore, it was reported that HPV induces breaks into cellular DNA and viral genome and that the preferential repair of lesions in viral episomes is critical for viral genome amplification [45]. It is possible that in HPV-infected cells, the reduction is p53 levels together with other alterations in factors controlling cell cycle caused by viral oncogenes may enhance the effects of spontaneous DNA damage leading to more deleterious lesions, such as DSBs. The ATM/CHK2/BRCA1 axis, as well as DSB repair, alleviates the effects of these lesions allowing the cells to survive. We can speculate that the silencing or inhibition of this pathway, in the absence of viral replication as occurs in HPV-transformed cells or cells transduced with HPV-16 E6, may result in the accumulation of DNA lesions that may trigger death signals leading to the reduction of cell survival. This is supported by our observation that ATM and CHK2 inhibition induces the accumulation of a sub-G1 cell population. Moreover, the interference of HPV oncoproteins in ATM/CHK2/BRCA1 signaling axis has been described by others [7,46]. In particular, it was observed that DNA damage may induce E6 phosphorylation through CHK1 and CHK2 increasing the ability of this viral oncoprotein to inhibit p53 transcriptional activity [47]. This is consistent with our observation that ATM, CHK2, and BRCA1 silencing/inhibition preferentially affect cells expressing wild-type E6 (Figure 7). Moreover, sensitivity to ATM, CHK2, and BRCA1 silencing/inhibition was reverted in cells expressing an E6 gene with mutations in a domain required for p53 degradation or cells where E6-AP was silenced. Interestingly, we analyzed different expression data series and showed that mRNA levels of CHEK2 and BRCA1 steadily increased from normal cervical tissue to CIN2, CIN3, and invasive squamous carcinoma samples. This observation suggests that upregulation of these factors plays a role in cervical cancer onset/progression and may explain, at least in part, the dependence of HPV-transformed cells on their continuous expression as shown in our study. We have previously demonstrated that cervical cancer-derived cell lines viability depends on specific components of the DNA repair machinery and DNA damage signaling [23,48]. Finally, we also show that treatment with chemotherapeutic drugs that induce DNA damage, such as cisplatin and doxorubicin, have their effect augmented when ATM kinase activity is inhibited. This effect is much more evident in cells depleted of p53 expression, such as HPV-positive cells and HPV-negative cells in which p53 is inactivated by mutation.

In the present study we show that cells expressing HPV E6 oncogene are highly dependent on the ATM/CHK2/BRCA1 signaling axis. Due to the direct involvement of this axis in DBSs repair, and the sensitization caused by doxorubicin and cisplatin, that cause DSBs, it is likely that this type of lesions and associated repair mechanisms play an important role in the effects described above. This is further supported by the fact that many genes identified in our screening as essential for HPV-transformed cells viability are involved in DSBs repair (Appendix A). Importantly, our results also point to an important role of p53, a central factor in dealing with DSBs and determining the fate of cells suffering DNA damage [49]. The higher sensitivity of HPV-transformed cells to ATM/CHK2/BRCA1 signaling axis inhibition, both in the presence and absence of DNA damage induction, may be explained by the fact that these cells divide at higher rates than normal PHKs. Therefore, we can assume that they are subjected to a higher degree of DNA replication stress and accumulation of DNA lesions. However, at this point we do not believe that differences in cell replication rate can fully explain our observations. For instance, we observed a similar dependence on ATM/CHK2/BRCA1 signaling axis in PHKs expressing only HPV16 E6. The doubling time of these cells is similar to that observed for normal PHKs (data nor shown). On the other hand, PHKs expressing only HPV-16 E7 proliferate at higher rates than normal PHKs. However, they are as resistant to ATM specific inhibition and to caffeine as normal PHKs (Figure 4C,D). Interestingly, the proliferation of these cells was affected by CHK2 inhibition, but to a lesser extent than that observed in cells expressing the oncogene E6. We are currently investigating the molecular mechanisms underlying the effects described above in our laboratory.

Our results may contribute to the improvement of therapeutic protocols for tumors with alterations in p53. In particular, for those associated with HPV infection. Altogether we present evidence that synthetic lethality using ATM/CHK2 inhibitors can be exploited to treat cervical cancer and other tumors in which p53 expression is inactivated by gene mutation or deletion.

## 4. Materials and Methods

### 4.1. Cell Lines and Retroviruses

Cervical cancer-derived cell lines SiHa (HPV16; ATCC^®^ HTB-35™, ATCC, Manassas, VA, USA), HeLa (HPV18; ATCC^®^ CCL-2™), and C33A (HPV-negative; ATCC^®^ CRM-HTB-31™) were cultured in MEM (Invitrogen, Thermo Fisher, Carlsbad, CA, USA) and HEK293T (ATCC^®^ CRL-3216™) and HaCaT cells were cultured in DMEM (Invitrogen), both supplemented with 10% FCS (Cultilab, Campinas, SP, Brazil) and maintained at 37 °C and 5% CO_2_. Primary Human Keratinocytes (PHK) (Lonza Walkersville, Inc., Walkersville, MD, USA) were grown in serum-free medium (Invitrogen) supplemented with recombinant epidermal growth factor (5 ng/mL) and bovine pituitary extract (50 mg/mL). PHK were transduced with recombinant retroviruses carrying the control vector (pLXSN) or vectors encoding HPV16 E6 and/or E7 or HPV16 E6 mutant, E6–8S9A10T, which cannot degrade p53 [50,51,52]. After 24 h, cells were selected with 300 µg/mL of G418 for 2 days, when 100% of mocked infected controls were dead. Surviving cells were expanded and used to seed monolayers cultures.

### 4.2. Library Manipulation and DNA Purification

Bacteria glycerol stocks of sequenced-verified MISSION^®^ shRNA lentiviral plasmids (Sigma-Aldrich, St. Louis, MO, USA) targeting tumor suppressors (SH0511) and DNA Repair Pathway (SH1811) were grown in the complex medium Luria Broth Medium (Sigma-Aldrich) with 75 µg/mL of ampicillin and incubated at 37 °C shaking at 200 rpm overnight. Plasmids were isolated with GenElute™ Plasmid Miniprep Kit (Sigma-Aldrich) and used to generate lentiviral virion production.

### 4.3. Lentiviral Virion Production and Quantification

A total of 1111 shRNA constructs cloned into pLKO1-puro plasmid backbone deficient viral vectors were cotransfected with the necessary helper plasmids for virus production (MISSION^®^ Lentiviral Packaging Mix, Sigma-Aldrich) using FuGENE^®^ HD Transfection Reagent (Promega, Madison, WI, USA) in HEK293T packaging cells. Culture medium was changed after 24 h, and supernatants containing lentivirus were collected after 48 and 72 h after transfection. Virus-containing supernatant was analyzed with an HIV-1 p24 antigen enzyme-linked immunosorbent assay (ELISA) kit (ZeptoMetrix Corporation, Buffalo, NY, USA) to estimate lentiviral vector titer.

### 4.4. Infection and Viability Assay

HeLa, SiHa, and PHK cell lines were seeded between 2000 and 3000 cells per well by using a 96-well format in a final volume of 100 µL per well. Twenty-four h after plating, cells were infected in triplicate with supernatants containing 10 MOI of lentivirus in the presence of 4 µg/mL hexadimethrine bromide (Polybrene; Sigma-Aldrich). Plates were centrifuged at 1200× *g* for 22 min. Media was replaced 24 h after infection and 2 μg/mL puromycin was added to one of the triplicate infected wells 72 h post infection. Viability was estimated five days after infection. For this, 10 µL of Alamar blue (Invitrogen) were added per well and cells were incubated at 37 °C for 4 h before absorbance measurement at 560 and 600 nm in in an Epoch Microplate Spectrophotometer (Bio-Tek, Winooski, VT, USA).

### 4.5. Cell Proliferation, Clonogenic and Anchorage Independent Growth Assays

Transduced cells were seeded in a density of 100 cells per well in 6-well plates. After two weeks in culture, clonogenic potential was evaluated. Colonies were fixed with 1% ethanol and stained with 0.5% crystal violet (Fisher, Waltham, MA, USA) in 10% ethanol. Cell proliferation was evaluated by cell counting. Briefly, transduced cells were seeded in a density of 1000 cells per well in 24-well plates. Cells were counted in triplicates on a hemocytometer from the first to the eighth day. For anchorage-independent growth analysis assay, 500 cells were suspended in 500 μL of D10 medium with 0.6% agarose. Cells were seeded in 24-well plates previously coated with the 1% agarose, and covered with D10 medium. After 30 days, colonies were stained with 50 μL of 5 mg/mL MTT (3-(4,5-150 dimethylthiazol-2-yl)-2,5-diphenyltetrazolium bromide) and counted.

### 4.6. Pharmacological Inhibition ATM and CHK2 and DNA Damage Induction

HeLa, SiHa, and PHK expressing E6 (PHKE6) and/or E7 (PHKE7) or E68S9A10T (PHKE6mut) were seeded in 96-wells plates (5.000 cells/well) and after 24 h cells were treated with 2 mM ATM/ATR inhibitor Caffeine, or 10 μM of the specific ATM inhibitor KU-55933, or with 100 μM of the specific CHK2 inhibitor 2-(4-(4-Chlorophenoxy)phenyl)-1H-benzimidazole-5-carboxamide hydrate (CHK2 Inhibitor II hydrate) in the presence of 1 μM doxorubicin and/or 1 μM of cisplatin. All inhibitors and DNA damage inducers were purchased from Sigma-Aldrich. All treatments were performed in octuplicates. After 72 h, cell viability was estimated by Alamar Blue (Invitrogen) assay as described above.

### 4.7. Cell Cycle Analysis

The effect of ATM and Chk2 inhibition on the progression of HPV-transformed cell lines and normal keratinocytes through the cell cycle was determined by flow cytometry. Basically, cells were treated and described above. After 72 h, supernatant and attached cells were harvested, washed with PBS and fixed with ethanol 70%. DNA was labeled with propidium iodine (20 μg/mL, PI, Sigma-Aldrich) in PBS with RNAse (200 μg/mL) and 0.1% Triton X100. Finally, at least 10,000 events per sample were acquired using a FACSCalibur (BD Biosciences, Franklin Lakes, NJ, USA). Data obtained were analyzed with FlowJo software (FlowJo Enterprise, Ashland, OR, USA).

### 4.8. Protein Extraction and Immunoblotting

Total protein extracts were obtained from sub-confluent cell cultures of the cell lines described above. Cell cultures were incubated with 500 µL of cold lysis buffer (150 mM NaCl, 50 mM Tris-HCl [pH 7.5], 0,5% NP-40) in the presence of protease inhibitors (CompleteTM, Roche Diagnostics, Basel, Switzerland) and phosphatase inhibitors (PhosSTOP, Sigma-Aldrich). Extracts were cleared from debris by centrifugation (10,000× *g* for 20 min) at 4 °C, transferred to fresh tubes and sored at −80 °C until use. Protein concentrations were determined using the Bio-Rad protein assay (Bio-Rad Laboratories, Hercules, CA, USA). Thirty μg of proteins were resolved by electrophoresis through sodium dodecyl sulfate (10 to 12%) polyacrylamide gels and transferred to polyvinylidene difluoride membranes (Amersham Pharmacia Biotech, Piscataway, NJ, USA). The membranes were blocked for 1 h in 5% nonfat milk, probed with primary antibodies against ATM (Ab78), ATM (phospho S1981) (Ab81292), CHK2 (Ab47433), γH2AX (ab22551) from Abcam (Cambridge, MA, UK); and tubulin (sc-25259) from Santa Cruz Biotechnology (Santa Cruz, CA, USA), according to the manufacturer’s instructions. Membranes were reprobed with horseradish peroxidase (HRP)-conjugated secondary antibodies and revealed using enhanced chemiluminescence procedures according to the manufacturer’s recommendations (Amersham Pharmacia Biotech).

### 4.9. In Silico Gene Expression Analysis

Global mRNA expression and clinical data from cervical samples named GSE39001, GSE52904, GSE63514, GSE138080 were retrieved from the Gene Expression Omnibus public online platform (GEO http://www.ncbi.nlm.nih.gov/geo, accessed on 1 October 2021). All datasets were investigated for possible outliers using the site https://graphpad.com/quickcalcs/grubbs1/, accessed on 1 October 2021.

### 4.10. Statistical Analysis

Data were analyzed using unpaired t, Mann Whitney, one way ANOVA or Kruskal–Wallis tests. All statistical analyses were performed with GraphPad Prism version 5.00 for Windows (GraphPad Software, La Jolla, California, USA). A *p*-value of less than 0.05 was considered statistically significant.

## Figures and Tables

**Figure 1 pathogens-11-00637-f001:**
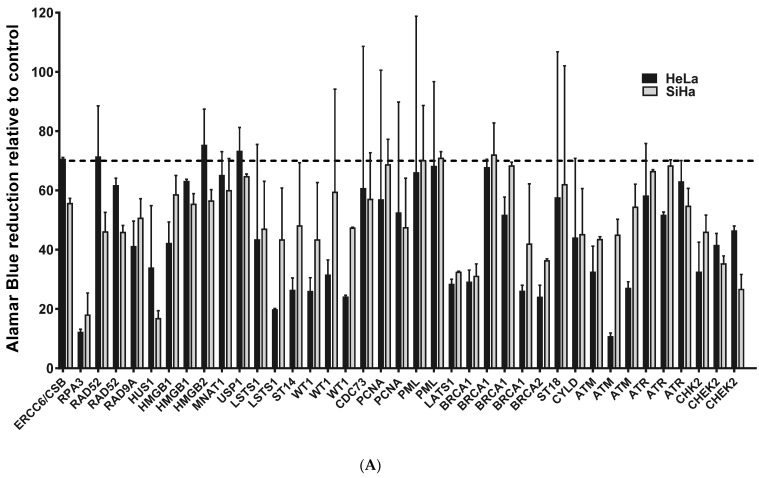
Screening for genes critical for cervical cancer-derived cell lines viability. A lentiviral shRNA library targeting 116 genes involved in DNA damage signaling/repair and tumor suppressors were used to silence specific genes in cervical cancer cell lines and PHK. Cells were seeded in 96 wells plates (2000 cells/well) and infected with lentiviral particles expressing specific shRNAs after 24 h. Alamar Blue was added (10 μL/well) after 72 h and its reduction was determined after four hours by absorbance measuring at 570 and 600 nm. Viability/proliferation inhibition values are presented as relative to those observed in control cells (each parental cell line transduced with scramble shRNA). (**A**) Candidate genes were arbitrary, defined as those which silenced reduced SiHa and HeLa cells proliferation by 30% or more. (**B**) Genes identified were further validated in PHK to determine those that induced inhibition of cell viability/proliferation preferentially in SiHa and HeLa cells (Student *t*-test *p* < 0.05 [*] for all comparisons between PHK and tumor cells).

**Figure 2 pathogens-11-00637-f002:**
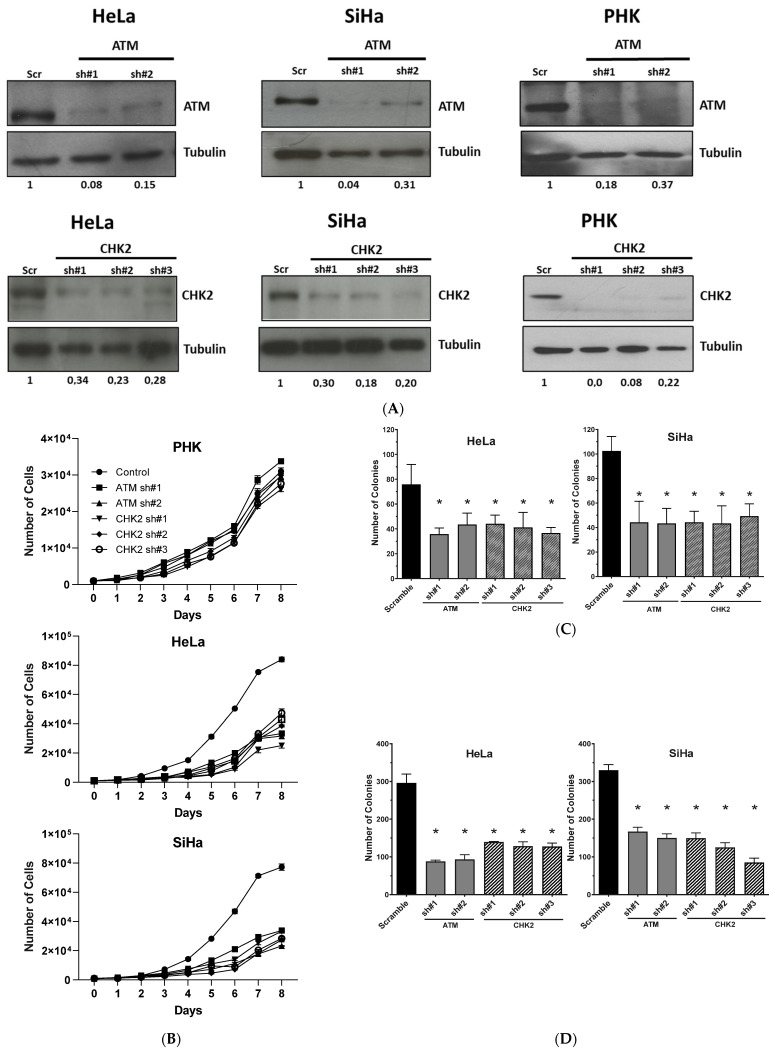
ATM and CHEK2 are essential for the proliferation and clonogenic potential of HPV-positive cervical cancer-derived cell lines. The expression of these genes in HeLa, SiHa, and PHK was silenced using specific shRNAs. (**A**) Gene silencing was confirmed by analyzing the levels of the encoded proteins by immunoblot. Cell growth curves, clonogenic and anchorage independent growth assays were performed to determine the effect of gene silencing on cell proliferation. (**B**) For growth curves cells were seeded in 24-wells plates (1000 cells/well) and counted daily for 8 days. For clonogenic potential analysis (**C**) cells were seeded in 6-wells plates (100 cells/well) and allowed to grow for two weeks, fixed, stained with crystal violet, and counted. For anchorage independent growth assays (**D**), cells were seeded in 24-wells plates (500 cells/well) and allowed to grow for 30 days. Colonies were stained with MTT and counted. Results presented are representative of three independent experiments (Student *t*-test *p* < 0.05 [*] for all comparisons between control and treated cells).

**Figure 3 pathogens-11-00637-f003:**
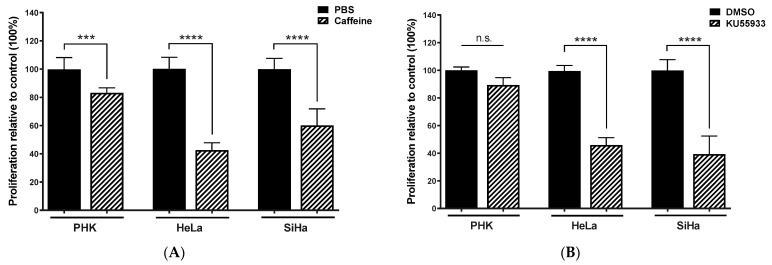
Inhibition of ATM and CHK2 reduces the proliferation/viability of HPV-positive cervical cancer-derived cell lines and induces the accumulation of sub-G1 cells. The activity of ATM and CHK2 kinases was inhibited using (**A**) caffeine (an inhibitor of ATM and ATR), (**B**) a specific inhibitor ATM and (**C**) a specific inhibitor of CHK2. Cells were seeded in 96 wells plates (5000 cells/well). Cells were treated with the inhibitors for 72 h and their proliferation was assessed by Alamar Blue reduction. (**D**) The distribution of cells populations in the different phases of the cell cycle was determined by flow cytometry. Results presented are representative of three independent experiments (Student *t*-test *p* < 0.05 [*], *p* < 0.01 [**], *p* < 0.001 [***], *p* < 0.0001 [****] for comparisons between control and treated cells).

**Figure 4 pathogens-11-00637-f004:**
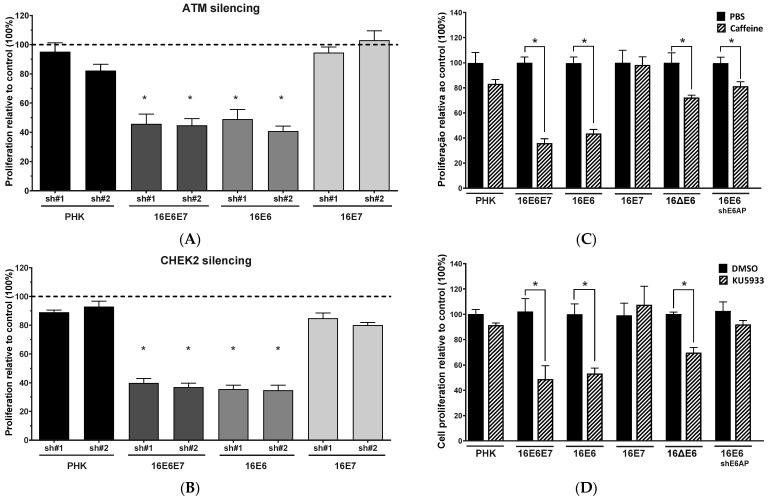
HPV16 E6 ability to induce p53 degradation is critical to sensitize primary human keratinocytes to ATM and CHEK2 silencing or kinase activity inhibition. PHK were transduced with retroviral vectors expressing wild type HPV-16 E6 and/or E7, or an HPV-16 mutant E6 unable to induce p53 degradation (16ΔE6). The expression of (**A**) ATM and (**B**) CHEK2 was silenced using specific shRNAs. Cells were seeded in 96-wells plates (5000 cells/well) and after 72 h their viability was assessed by Alamar Blue reduction. The activity of ATM and CHK2 kinases was inhibited using (**C**) caffeine (an inhibitor of ATM and ATR), (**D**) a specific inhibitor ATM and (**E**) a specific inhibitor of CHK2. Cells were seeded in 96-wells plates (5000 cells/well). Cells were treated with the inhibitors for 72 h and their viability was assessed by Alamar Blue reduction. The proliferation of PHK expressing HPV-16 E6 and transduced with a shRNA against the ubiquitin ligase responsible for p53 degradation in the presence of E6 (E6-AP) was not strongly affected by ATM or CHK2 inhibition (**C**–**E**). Results presented are representative of three independent experiments (Student *t*-test *p* < 0.05 [*] for all comparisons between control and treated cells).

**Figure 5 pathogens-11-00637-f005:**
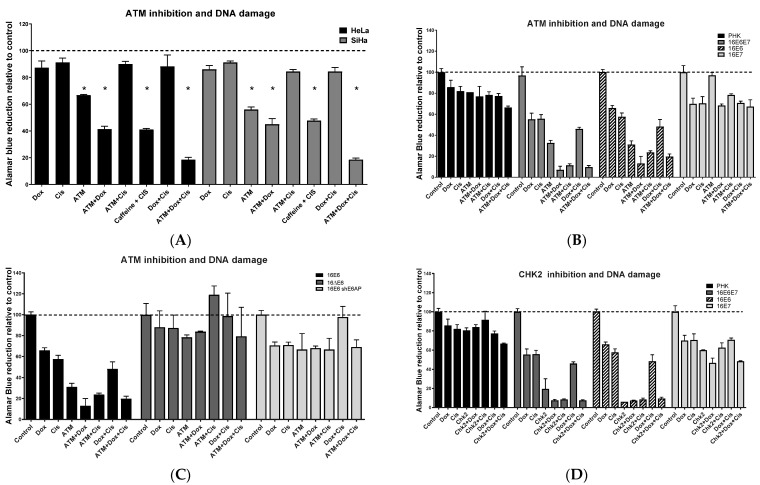
ATM and CHK2 inhibition sensitize cells expressing high-risk HPV E6 to DNA damage. (**A**) HeLa, SiHa cells; and PHK transduced with retroviral vectors expressing wild type (**B**,**D**) HPV-16 E6 and/or E7; (**C**,**E**) an HPV-16 mutant E6 unable to induce p53 degradation (16ΔE6); (**F**) and HPV-negative cells harboring mutations in p53 gene were seeded in 96 wells plates (5000 cells/well). The activity of ATM and CHK2 kinases was inhibited using caffeine (an inhibitor of ATM and ATR), a specific inhibitor ATM and a specific inhibitor of CHK2 in the presence of doxorubicin (1 μM) or cisplatin (1 μM). Cells were seeded in 96-wells plates (5000 cells/well). Cells were treated with the inhibitors for 72 h and their viability was assessed by Alamar Blue reduction. Results presented are representative of three independent experiments (Student *t*-test *p* < 0.05 [*] for all comparisons between control and treated cells).

**Figure 6 pathogens-11-00637-f006:**
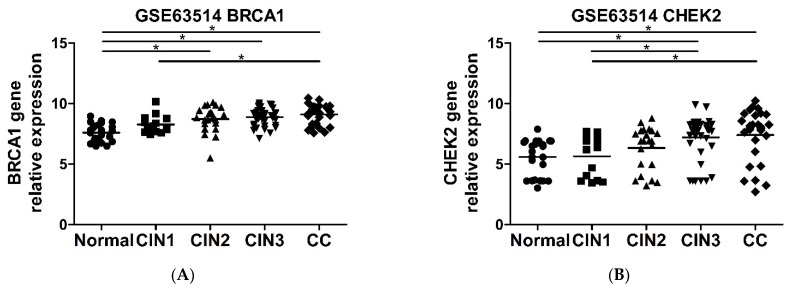
BRCA1 and CHEK2 mRNA expression is upregulated in cervical intraepithelial neoplasias (CIN) and invasive carcinomas. BRCA1 (**A**,**C**,**E**,**G**) and CHEK2 (**B**,**D**,**F**,**H**) gene expression is upregulated in CIN3 and cervical cancer (CC) when compared with normal cervical tissues. (**A**,**B**) GSE63514: Normal–n = 24; CIN1–n = 14; CIN2–n = 22; CIN3–n = 40; CC–n = 28. (**C**,**D**) GSE138080: Normal–n = 10; CIN–n = 15; CC–n = 10. (**E**,**F**) GSE52904: Normal–n = 17; CC–n = 55. (**G**,**H**) GSE39001: Normal–n = 5; CC–n = 19. *p* < 0.05 [*] by 1 way ANOVA (**A**,**D**), Kruskal–Wallis (**B**,**C**), unpaired *t* test (**H**) or Mann Whitney (**E**–**G**) test.

**Figure 7 pathogens-11-00637-f007:**
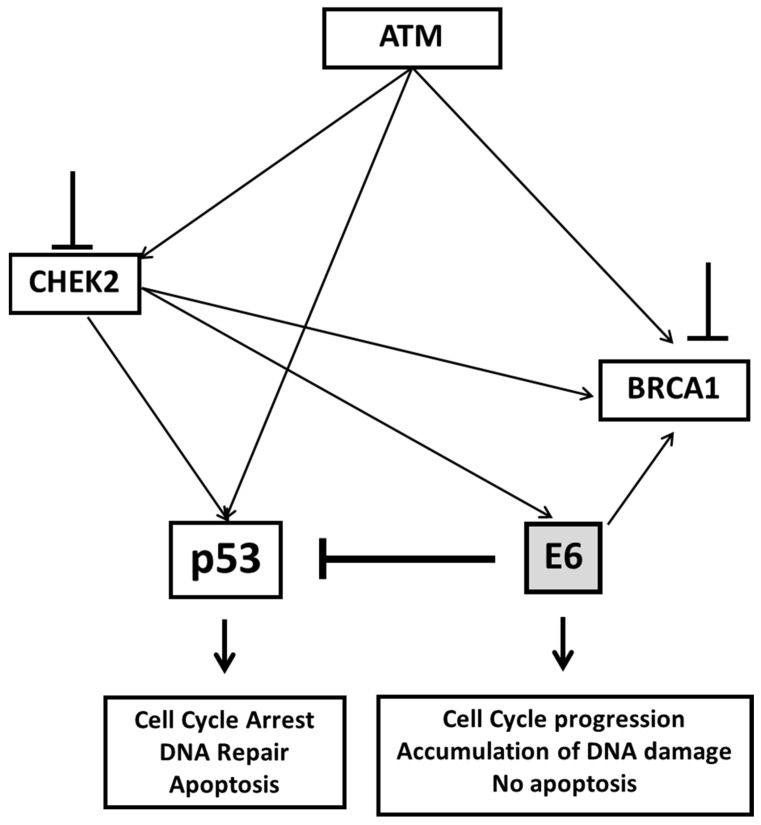
E6 expressing cells depend on ATM/CHK2/BRCA1 axis for survival. HPV E6 induces p53 degradation avoiding cell cycle arrest and apoptosis. Double strand breaks are processed by BRCA1 and other proteins warranting minimal genome stability. CHK2 signaling reinforces E6 inhibitory effect on p53. Silencing or inhibition of ATM/CHK2/BRCA1 shifts the balance inducing the accumulation of DNA lesions that reduce cell viability.

## Data Availability

The datasets used and/or analyzed during the current study are available from the corresponding author on reasonable request. Links to publicly archived datasets analyzed during the study: GSE39001: https://www.ncbi.nlm.nih.gov/geo/query/acc.cgi?acc=GSE39001; GSE52904: https://www.ncbi.nlm.nih.gov/geo/query/acc.cgi?acc=GSE52904; GSE63514: https://www.ncbi.nlm.nih.gov/geo/query/acc.cgi?acc=GSE63514; GSE138080: https://www.ncbi.nlm.nih.gov/geo/query/acc.cgi?acc=GSE138080, accessed on 1 October 2021.

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
