# Peer review of "ATM Pathway Is Essential for HPV–Positive Human Cervical Cancer-Derived Cell Lines Viability and Proliferation"

_pathogens, 2022, doi:10.3390/pathogens11060637_

Round 1

Reviewer 1 Report

In this manuscript, Abjaude et al. have addressed some of my previous comments. However, they have still not addressedthe more important issues I had with the original submission.

  • Figure 3A-D - western blot data showing efficacy of the inhibitors used must be shown in each cell line. The authors have only added data for KU-55933, but not for caffeine or the CHK2 inhibitor. This is should be done for correct interpretation of the data.
  • A bigger issues is in Figure 4 - the authors have still not provided data showing expression of HPV oncoproteins in PHK cells or p53 expression in cells expression16E6ΔE6 or shE6AP treated cells. This data MUST be provided.

Reviewer 2 Report

The submitted paper “ATM PATHWAY IS ESSENTIAL FOR HPV-POSITIVE HUMAN CERVICAL CANCER-DERIVED CELL LINES VIABILITY AND PROLIFERATION” by Abjaude et al. seeks to establish that synthetic lethality is playing a role in the differential response of HPV+ cervical cancer cell lines and primary human keratinocytes transduced with HPV oncogenes.  The paper presents a mixed bag of observations.  The experiments appear to be well conducted, for the most part, but the conclusions are somewhat overstated and confounding. 

In particular the following issues need to be addressed:

--The relative proliferation of SiHa and HeLa cells compared to PHKs is not directly addressed and is likely responsible for the differential effects of the ATM/Chk2 inhibition.  The cancer cells are dividing much more rapidly and therefore subject to a high degree of DNA replication stress, which is not even mentioned in the paper.  This is a major shortcoming.

--Along this same line, it is well known that cisplatin and doxorubicin cause dsDNA breaks in proliferating cells.  Therefore it is not surprising that inhibiting dsDNA repair pathways, including elements such as ATM and Chk2, results in enhanced cell death.  The paper avoids discussing the accumulation of dsDNA breaks and would be improved had some data been including such as gamma-H2AX labelling or some other indicator of dsDNA breaks, which of course are expected to accumulate faster in cells under replication stress.  The paper doesn’t even mention that both cisplatin and doxorubicin are well known to act via introduction of dsDNA breaks when introducing these drugs at the top of page 11.  Likewise, the role p53 in response to dsDNA breaks introduced by chemotherapy is well known and supported by a cast literature that is largely avoided in the paper.  In fact, the paper mentions dsDNA breaks exactly ONE TIME in the discussion.  How can this be?

--The words viability, proliferation, and survival are carelessly used almost interchangeably throughout the paper.  The paper could be more thoughtful with its choice of words.

--The concept of synthetic lethality regarding p53 and the ATM pathway is interesting but it isn’t convincing due to the large number of complicating factors and variables at play in the paper.  In addition, the paper misstates a number of facts regarding the current status of understanding with regards to ATM control of HPV replication.  For instance, the paper states that “”Interestingly, recent studies showed that proteins phosphorylated by ATM, such as BRCA1, RAD51 and CHK2, are essentials to HPV genome amplification and maintenance [7,43].“”  It’s true that these papers have implicated these elements in productive replication, but not maintenance replication.  In fact, it’s primarily the ATR pathway has been implicated as important for maintenance.  See, for example:  Edwards et al., J. Virol., 87 (2013), pp. 3979-3989, and Hong et al., mBio 6:e02006-15.

Reviewer 3 Report

The authors present convincing evidence that inhibition of the ATM/CHK2 pathway may be an option to target cervical cancers. The evidence is based on lentiviral knockdown of both ATM and CHEK2 in 2 cervical cancer cell lines (genes chosen by using shRNA screen  of genes involved in DNA damage), and validated using chemical inhibition of the gene encoded kinases. They go on to show that the ATM/CHK2 pathway is important for  the viability of cells (Alamar Blue Assay, cell growth curves, cell cycle analysis) likely in a manner which involves the p53 tumor suppressor. Cells which express HPV E6 able to target p53 for degradation, or have mutant p53 (HaCat, C33A) have enhanced sensitivity to ATM/CHK2 inhibition. Cell cycle analysis suggests that the viability reduction (as per Alamar blue assay) is due in part to apoptosis, as evidenced by increase in subG1 population. However the exact mechanism of viability reduction (apoptosis and/or reduction of proliferation) cannot be accurately deduced from the assays used. The authors should more clearly discuss this (or provide any relevant data which may clarify).  Another point for discussion would be the sensitivity of E7 expressing PHK to CHK2 inhibition (Figure 4E). How do the authors interpret this? 

While the overall language of the paper is of high quality this reviewer has identified some minor points which require spell check.

line 31: "evidence"

line 32: "HPV"

line 61: "of which"

Figure 3D labels: "Caffeine"

Round 2

Reviewer 1 Report

Whilst I appreciate that the authors have extensively published manuscripts utilising the plasmids/vectors used in this manuscript, it is still my opinion that for any new manuscript, authors must demonstrate the correct expression/ function of these plasmids/vectors/knockdowns for the reader to assess, particularly if the reader has not read some of the other publications from the group.

This is particularly true in this manuscript, due to the use of different cellular systems, as the authors state in their reply: 'We can present some of these controls to Reviewer #1, although they were not conducted with the exact same cells that those used in the present study (the packaging cells producing the amphotropic retroviruses expressing HPV16 oncogenes are just the same that those used in our study).'

It is my opinion that the authors must demonstrate correct expression of all expression vectors or knockdown constructs used in the cellular system used in the current manuscript.

Author Response

R1: “Whilst I appreciate that the authors have extensively published manuscripts utilising the plasmids/vectors used in this manuscript, it is still my opinion that for any new manuscript, authors must demonstrate the correct expression/ function of these plasmids/vectors/knockdowns for the reader to assess, particularly if the reader has not read some of the other publications from the group.” 

In the present version of the manuscript, we have included information (Supplementary Figure S7) that confirms the correct expression/function of the plasmids/vectors/knockdowns and inhibitors used in our study. It is really difficult to provide all the information needed by the reader that has not read other publications from our group or other groups. However, we believe that the information presented here is enough to the correct interpretation to the results of the study.

 R1: “This is particularly true in this manuscript, due to the use of different cellular systems, as the authors state in their reply: 'We can present some of these controls to Reviewer #1, although they were not conducted with the exact same cells that those used in the present study (the packaging cells producing the amphotropic retroviruses expressing HPV16 oncogenes are just the same that those used in our study).'.” 

Here, we present consistent data proving the expression and function of the HPV oncogenes in primary human keratinocytes (PHKs). The cellular systems used are worldwide used cervical cancer derived cell lines and commercially available PHKs. The confirmation of expression and function of HPV-16 oncogenes was performed, as our routine laboratory controls, in PHKs provided by LONZA. These cells may come from different vials and keratinocytes lots than those used to perform the specific experiments presented in our study. Therefore, although pooled, we cannot swear that they are exactly cells from the same combination of donors. Besides, our experiments were performed with different PHKs vials. To perform all the controls required by Reviewer #1 for each vial used would be an impossible task.

R1: “It is my opinion that the authors must demonstrate correct expression of all expression vectors or knockdown constructs used in the cellular system used in the current manuscript.

With the inclusion of the new Supplementary Figure S7 we believe that all the information required by Reviewer #1 has been provided.   

This manuscript is a resubmission of an earlier submission. The following is a list of the peer review reports and author responses from that submission.

Round 1

Reviewer 1 Report

The authors put forth the argument that inhibition of the ATM-Chk2 axis may provide a synthetic lethal approach for treatment of cervical cancers in which the viral E6 gene targets the host cell p53 for degradation.  The literature has many examples of synthetic lethality between p53 and ATM, but the focus upon HPV brings some degree of novelty.  

However, the paper is plague by assumptions with regards to specificity of results that don’t appear to be warranted:

--Caffeine inhibits many cellular components beyond ATM (including ATR) but the paper assumes its effects are mediated via ATM.

--E6 mutants are discussed that are assumed to specifically bypass p53 inhibition/degradation, but it’s likely that the mutations result in E6 misfolding affecting many E6 functions beyond p53 degradation (see:  Mol. Cell (2006) 21(5): 665-678).  Therefore conclusions regarding p53 specificity are questionable

The paper compares results with the cancer cell lines HeLa and SiHa to primary human keratinocytes (PHKs), but the cell growth conditions are very different. PHKs are cultured without serum while cancer lines are in 10% FBS and therefore exposed to a wealth of factors not available to .  Therefore the proliferation kinetics, levels of replication stress, and resulting activation of DDR will be quite different and likely contribute to differences seen between PHK and cancer lines.  

Along this same line, it’s well established in the literature that the HPV oncogenes cause DDR activation in cells.  It’s therefore important to directly compare levels of DDR genes in PHK to levels in HeLa and SiHa, and this is not done.

A potential source of confusion in the paper is the failure to distinguish between studies of cells carrying HPV integrated genomes in cancer cells (as used in this paper) and studies examining the replication and maintenance of extrachromosomal HPV genomes (episomes) in non-cancerous cells (references 9, 10, 35, and 36).  Little is done to distinguish between these two systems that are regulated very differently.

HaCat and C33 cell p53 status needs to be clarified.  The paper states:  ""We also observed that C33 cell lines, in which p53 is not detected due to gene deletion, had viability dropping to 20% after combined treatment of ATM inhibitor with doxorubicin and cisplatin.""  There are some problems here: the authors indicate C33 cells have a p53R273C mutation in text and yet claim there is no p53 because of deletion.  Such statements do not raise confidence.  The authors don't describe C33 cells in Materials and Methods section, while the ATCC indicates that C33A cells over-express p53.   Furthermore p53(R273C) is a gain of function mutation.  How does this compare with p53 null cells or with cells expressing HPV E6?  Are C33 cells the same as C33A cells? 

The observations summarized in Fig. 5 are confounding. The drug treatment studies conducted with Doxorubicin and Cisplatin are problematic.  Low doses of both drugs are used that have only minimal adverse effects upon cell proliferation/viability as measured by Alomar blue assay.  ED50 or LD50 or some quantifiable, routine measure of cytotoxic drug effect on various cells is missing.  Isn’t it likely that the Alomar blue IC50 of dox or cis (or some other measure of degree of potency) against PHK is different than in HeLa or SiHa or other cell types/conditions?  How does one effectively make comparisons between various cells/conditions without this information?  A much more interesting, accurate, and helpful approach would be to calculate the effects of inhibition of ATM, CHEK2, and other parameters on IC50 under various conditions.  Otherwise the results become a phenomenological quagmire.

Finally, the GEO mRNA expression in silico data do little to advance the central points of the paper.

Other points: English language usage is adequate but awkward and could benefit from a good editing.

Anchorage-independent growth assay not in M&M

Reviewer 2 Report

In this manuscript, Abjaude et al. demonstrate the role of ATM signalling in HPV+ cervical cancer. Whilst potentially interesting, the current version of the manuscript contain several issues pertaining to both the performed experiments and the writing of the manuscript itself. At this point, the manuscript in not in a publishable form - please see specific comments below.

  • The references in the introduction are inadequate. Many of them are old references (e.g. Narisawa-Saito et al, Cancer Sci., 2007 on the mechanism of E6/E7 function. This should be an up to date refence on the function of these proteins, such as Scarth et al., JGV, 2021.). Furthermore, replication of HPV via the E1/E2 complex also induces DNA damage - important references from the lab of Iain Morgan and Alison McBride are missing and should be included for a complete understand of the topic at hand

  • line 116 - 'The silencing of all other 18 genes reduced the viability of both normal PHK and cervical cancer derived cell lines (data not shown).' This data could easily be shown as a supplemental figure - please add

  • Figure 2A - the blot for ATM/Tubulin in PHKs is of poor quality and should be replaced with a higher quality blot. Additionally, it is not acceptable to have the scramble control blot for CHK2 in PHKs be from a separate blot - this must be run on the same gel as the CHK shRNA samples
  • Figure 2C-D - representative images of (at least) the anchorage dependent colony formation should be provide as supplementary data

  • Figure 3A-D - western blot data showing efficacy of the inhibitors used must be shown in each cell line. Please also added inhibitor concentration to figure legend as well as methods

  • Figure 4 - western blot data showing expression of HPV oncoproteins in PHK cells, p53 expression and there for lack of function in 16E6ΔE6 and shE6AP treated cells, and E6AP knockdown efficiency must be provided